# Jitter-Robust Phase Retrieval Wavefront Sensing Algorithms

**DOI:** 10.3390/s22155584

**Published:** 2022-07-26

**Authors:** Liang Guo, Guohao Ju, Boqian Xu, Xiaoquan Bai, Qingyu Meng, Fengyi Jiang, Shuyan Xu

**Affiliations:** 1Changchun Institute of Optics, Fine Mechanics and Physics, Chinese Academy of Sciences, Changchun 130033, China; guoliang18@mails.ucas.ac.cn (L.G.); ciomp_xubq@126.com (B.X.); xiaoquanbai93@163.com (X.B.); mengqy@ciomp.ac.cn (Q.M.); jiangfengyi@ciomp.ac.cn (F.J.); xusy@ciomp.ac.cn (S.X.); 2University of Chinese Academy of Sciences, Beijing 100049, China; 3Chinese Academy of Sciences Key Laboratory of On-Orbit Manufacturing and Integration for Space Optics System, Changchun 130033, China

**Keywords:** jitter-robust, parametric phase retrieval, iterative-transform wavefront sensing

## Abstract

Phase retrieval wavefront sensing methods are now of importance for imaging quality maintenance of space telescopes. However, their accuracy is susceptible to line-of-sight jitter due to the micro-vibration of the platform, which changes the intensity distribution of the image. The effect of the jitter shows some stochastic properties and it is hard to present an analytic solution to this problem. This paper establishes a framework for jitter-robust image-based wavefront sensing algorithm, which utilizes two-dimensional Gaussian convolution to describe the effect of jitter on an image. On this basis, two classes of jitter-robust phase retrieval algorithms are proposed, which can be categorized into iterative-transform algorithms and parametric algorithms, respectively. Further discussions are presented for the cases where the magnitude of jitter is unknown to us. Detailed simulations and a real experiment are performed to demonstrate the effectiveness and practicality of the proposed approaches. This work improves the accuracy and practicality of the phase retrieval wavefront sensing methods in the space condition with non-ignorable micro-vibration.

## 1. Introduction

The imaging performance of space-based large-aperture astronomical telescopes is susceptible to space thermal/mechanical perturbations, which can gradually lead to mirror misalignments or deformations during long-term on-orbit observation and operation [1]. Mirror misalignments and deformations will introduce wavefront aberrations and degrade the imaging quality of the system. In order to maintain the imaging quality, it is of critical importance to accurately sense the wavefront of the system. On this basis, we can actively align the system and correct those aberrations with mirror actuators [2].

Image-based wavefront sensing (WFS) represents a class of WFS methods that directly utilize image plane intensity measurements to recover the pupil plane wavefront phase of the optical system. This class of methods mainly includes iterative-transform methods (developed from the Gerchberg-Saxton algorithm) [3,4,5,6,7], parametric methods (also known as model-based optimization algorithms or directly called phase diversity algorithms) [8,9,10,11,12], and deep learning methods [13,14,15,16,17,18,19,20]. Compared to other WFS methods, such as the Shack–Hartmann sensor [21], pyramid sensor [22], or curvature sensing [23], this class of WFS methods is particularly suitable for space applications [24,25].

However, the accuracy of the image-based wavefront sensing methods for space telescopes is inevitably influenced by space perturbations. Meanwhile, line-of-sight jitter is one of the most common forms of disturbances due to micro-vibration of the platform which can be induced by a refrigerator or reaction wheel [26]. Line-of-sight jitter can change the intensity distribution in the image plane and therefore reduce the accuracy of the recovered wavefront phase. In a sense, the effect of jitter is hard to correct. On one hand, it is particularly difficult to eliminate the micro-vibration of the platform which usually has a wide frequency range; on the other hand, the effect of jitter on intensity distribution shows some stochastic properties and it is hard to present an analytic model to describe it. Consequently, seldom researches can present an effective solution to this problem.

Many researchers have studied how to reduce the effect of jitter on imaging quality. Sun et al. proposed a new mathematical method named DIROEF to estimate the true value of the jitter, and the SVBM and VACM methods were used to compensate for the radiometric and geometric effects of the jitter on the image [27]. Liu et al. presented an approach that offers jitter effect compensation and quality improvement for push-broom optical images based on dynamic point spread function (PSF) estimation and iterative image restoration [28,29]. Wu et al. presented an algorithm using Fourier analysis to resolve the jitter function for a HiRISE image that is then used to update instrument pointing information to remove geometric distortions from the image [30]. However, few researchers study how to reduce the effect of jitter on the accuracy of image-based wavefront sensing, which is of importance for space applications.

While the effect of jitter presents stochastic property, it still obeys some statistics laws. In the presence of jitter, the position of the intersection of the line-of-sight with the image plane at any moment within the exposure time obeys normal distribution. Based on this fact, Gaussian convolution is introduced to describe the effect of jitter on image [31,32]. Then this paper proposes two jitter-robust image-based wavefront sensing algorithms, which can be categorized into iterative-transform algorithms and parametric algorithms, respectively. The parametric jitter-robust phase retrieval algorithm is also extended to those cases where the magnitude of jitter is not known to us. Detailed simulations and a real experiment are performed to demonstrate the effectiveness and practicality of the proposed approach.

The remainder of this paper is organized as follows. In Section 2, we propose the principle of parametric phase retrieval wavefront sensing algorithm which takes the effects of jitter into consideration. Simulations and a real experiment are conducted in Section 3 and Section 4, respectively. In Section 5, we summarize and conclude the paper.

## 2. Principle of Jitter-Robust Phase Retrieval Wavefront Sensing Algorithm

This section first discusses how to describe the intensity distribution of the point spread function (PSF) in the presence of jitter. On this basis, two kinds of jitter-robust image-based wavefront sensing algorithms are proposed, which can be categorized into iterative-transform algorithms and parametric algorithms, respectively.

Specifically, the first jitter-robust algorithm involves iterative Fourier transformation back and forth between the object and Fourier domains and application of the measured data and established imaging model in the presence of jitter. The second jitter-robust algorithm recovers the parameterized wavefront aberrations by minimizing the objective function which includes the effects of jitter with nonlinear optimization methods.

### 2.1. Modelling Point Spread Function in the Presence of Jitter

We will first discuss how to describe the intensity distribution of the point spread function (PSF) in the presence of jitter. Let us suppose that the object is illuminated with noncoherent quasi-monochromatic light, and the imaging system is a linear shift-invariant system. According to Fourier optics, the point spread function in the image plane can be expressed as
(1)$$I\left(u\right)={\left\{FT^{-1}\left\{Aexp\left[i\emptyset \left(x\right)\right]\right\}\right\}}^{2}$$
where I represents the intensity distribution of a PSF, which is a two-dimensional matrix, i is an imaginary unit, $FT^{-1}$ represents the inverse Fourier transform, A is a binary two-dimensional matrix, and the value of the matrix element is 1 inside the normalized aperture and the rest is 0, and x represents the aberration coefficients vector used to describe the wavefront phase map.

While the effect of jitter presents stochastic property, it still obeys some statistics laws. In the presence of jitter, the position of the intersection of the line-of-sight with the image plane at any moment within the exposure time obeys normal distribution. Therefore, Gaussian convolution can be introduced to describe the effect of jitter on the image during the exposure time. In other words, in the presence of jitter, the intensity distribution of a PSF can be modeled as:(2)$$I\left(u\right)={\left\{FT^{-1}\left\{Aexp\left[i\emptyset \left(x\right)\right]\right\}\right\}}^{2}⊗G$$
where *G* is a Gaussian convolution kernel, which is used to represent the effect of jitter on the intensity distribution of the PSF. To accurately describe the effect of jitter, the element in *G* should be carefully determined.

The value of the element in *G* can be determined according to the two-dimensional Gaussian normal distribution, which can be expressed as
(3)$$g\left(u,v;\sigma_{u},\sigma_{v},\rho_{uv}\right)=\frac{1}{2\pi \sigma_{u}\sigma_{v}\sqrt{1-\rho_{uv}^{2}}}\exp\left\{\frac{-1}{2\left(1-\rho_{uv}^{2}\right)}\left[\frac{u^{2}}{\sigma_{u}^{2}}-\frac{2\rho_{uv}uv}{\sigma_{u}\sigma_{v}}+\frac{v^{2}}{\sigma_{v}^{2}}\right]\right\}$$
where *u* and *v* are coordinate variables (can also be seen as the values of two random variables), $\sigma_{u}$ and $\sigma_{v}$ represent the variances of the two random variables, respectively, and $\rho_{uv}\;$ denotes the correlation coefficient between the two random variables. $\sigma_{u}$,$\;\sigma_{v}$ and $\rho_{uv}$ determines the distribution form of the two-dimensional Gaussian function. In this Equation, we suppose the mean values of the two random variables are zeros. Here G and g have different meanings. *G* is the Gaussian kernel which is used to perform convolution and g is the Gaussian function which is used to discuss the property of the Gaussian function.

In general, we can further suppose$\;\sigma_{x}$ =$\sigma_{y}$, and $\rho_{xy}=0$. In this case, the Gaussian function in Equation (3) can be rewritten as
(4)$$g\left(x,y;\sigma \right)=\frac{1}{2\pi \sigma^{2}}\exp\left\{-\frac{x^{2}+y^{2}}{2\sigma^{2}}\right\}$$

The value of σ in this equation should be consistent with the variance of line-of-sight jitter measured in pixels.

On this basis, we can obtain the value of each element in the Gaussian kernel, G. Specifically, we can first draw grid lines in a plane, and the intervals between grid lines are equivalent to the pixel size. Then, we assume the center of the Gaussian function is in consistent with the center of the central square in the grid, as shown in Figure 1, and calculate the weighted sum of the Gaussian function in each square in the grid through integration in each square. The number of the squares in the grid (i.e., the number of the elements in the Gaussian kernel) is usually 3 × 3 or 5 × 5. Note that the sum of element values in the Gaussian kernel should be normalized to 1.

### 2.2. Iterative-Transform Jitter-Robust Image-Based Wavefront Sensing Algorithm

According to the convolution theorem, Equation (2) can be rewritten as
(5)$$FT\left\{I\left(u\right)\right\}=FT\left\{{\left\{FT^{-1}\left\{\mathrm{Aexp}\left[i\emptyset \left(x\right)\right]\right\}\right\}}^{2}\right\}\cdot FT\left(G\right)$$

This equation can further be rewritten as
(6)$$FT^{-1}\left\{\frac{FT\left\{I\left(u\right)\right\}}{FT\left\{G\right\}+\gamma }\right\}={\left\{FT^{-1}\left\{Aexp\left[i\emptyset \left(x\right)\right]\right\}\right\}}^{2}$$
where γ is introduced to suppress the effect of noise near the nulls of the $\frac{FT\left\{I\left(u\right)\right\}}{FT\left\{G\right\}}$.

On this basis, we can propose an iterative-transform jitter-robust image-based wavefront sensing algorithm. When only one focal image is available, this algorithm is illustrated in Figure 2. In this figure, θ is the phase map of the complex field after Fourier transformation. This phase map can be represented by y which is a vector including a set of Zernike coefficients. k is the iteration index. The key point in Figure 2 is that $\sqrt{I\left(u\right)}$ is replaced by $\sqrt{FT^{-1}\left\{\frac{FT\left\{I\left(u\right)\right\}}{FT\left\{G\right\}+\gamma }\right\}}$.

The algorithm illustrated in Figure 2 can easily be extended to those cases where two PSFs with different defocused distances are available.

### 2.3. Parametric Jitter-Robust Image-Based Wavefront Sensing Algorithm

Since the relationship between PSF and aberration coefficients in the presence of jitter has been established, we can present the following equations after collecting two PSFs with a detector,
(7)$$\left\{\begin{array}{c} I_{1}={\left\{FT^{-1}\left\{Aexp\left[i\emptyset \left(x\right)\right]\right\}\right\}}^{2}⊗G\\ I_{2}={\left\{FT^{-1}\left\{Aexp\left[i\emptyset \left(x+∆\right)\right]\right\}\right\}}^{2}⊗G \end{array}\right.$$
where $I_{1}$ and $I_{2}$ are the intensity distributions of the two PSFs collected with detector, ∆ is the known diversity phase between the two PSFs.

It is hard to directly obtain the analytical solution of the complicated nonlinear equations (i.e., x) presented above (both $I_{1}$ and $I_{2}$ are matrices and therefore there can be hundreds of equations in Equation (7)). In order to solve the aberration coefficients from these equations, the following objective function is first established,
(8)$$E\left(x\right)=\sum {\left|I_{1}-\left\{FT^{-1}\left\{Aexp\left[i\emptyset \left(x\right)\right]\right\}\right\}⊗G\right|}^{2}\;+\sum {\left|I_{2}-\left\{FT^{-1}\left\{Aexp\left[i\emptyset \left(x+∆\right)\right]\right\}\right\}⊗G\right|}^{2}$$
where $\sum \left(\cdot \right)$ means the integral over all the effective pixels in the two PSF images.

Optimization methods can then be used to find the minimum of $E\left(x\right)$. Under ideal conditions, $E\left(x_{0}\right)$ is equal to zero if $x_{0}$ is the solution of the equations. In practice, $E\left(x\right)$ cannot reach zero due to image noise and we generally consider $x_{0}{}^{\prime }$ is the solution of the equations when $E\left(x_{0}{}^{\prime }\right)$ is the minimum of $E\left(x\right)$. The diagram of this parametric algorithm is shown in Figure 3.

Here we will derive the analytic partial derivative of E with respect to an aberration coefficient. Supposing the aberration coefficients vector x is expressed as
(9)$$x=\left[\alpha_{1,}\alpha_{2},\alpha_{3},\cdots \alpha_{N}\right]$$
where N is total aberration coefficients considered in the algorithm, then the phase $\emptyset \left(x\right)$ can be represented as
(10)$$\emptyset \left(x\right)=\overset{N}{\underset{j=1}{\sum }}\alpha_{j}Z_{j}$$
where$\;Z_{j}$ is the *j*th Fringe Zernike coefficients [33].

According to the convolution derivation principle [34],
(11)$$\frac{\partial F⊗G}{\partial x}=\frac{\partial F}{\partial x}⊗G$$
where F represents a function with variable x, then the partial derivative of E with respect to an aberration coefficient $\alpha_{j}$ can be derived, which is presented in Equation (12),
(12)$$\frac{\partial E}{\partial \alpha_{j}}=4\cdot \sum \left[I_{1}-{\left|F_{1}\left(x\right)\right|}^{2}⊗G\right]\cdot \left|F_{1}\left(x\right)\right|\cdot FT^{-1}\left\{f_{1}\left(x\right)\cdot i\cdot Z_{j}\right\}⊗G\;+4\cdot \sum \left[I_{2}-{\left|F_{2}\left(x\right)\right|}^{2}⊗G\right]\cdot \left|F_{2}\left(x\right)\right|\cdot FT^{-1}\left\{f_{2}\left(x\right)\cdot i\cdot Z_{j}\right\}⊗G$$
where
(13)$$\left\{\begin{array}{c} F_{1}\left(u\right)=FT^{-1}\left\{Aexp\left[i\emptyset \left(x\right)\right]\right\}\\ F_{2}\left(u\right)=FT^{-1}\left\{Aexp\left[i\emptyset \left(x+∆\right)\right]\right\} \end{array}\right.$$
and
(14)$$\left\{\begin{array}{c} f_{1}\left(x\right)=Aexp\left[i\emptyset \left(x\right)\right]\\ f_{2}\left(x\right)=Aexp\left[i\emptyset \left(x+∆\right)\right] \end{array}\right.$$

The $\sum \left(\cdot \right)\;$in Equation (12) means integration over the effective region of the focal image. This analytic form for the partial derivative of E with respect to an aberration coefficient can be used with any gradient-search optimization algorithm, such as Gradient Descent, Quasi-Newton, Conjugate Gradient or BFGS algorithm (BFGS is used in this work).

## 3. Simulations

### 3.1. Simulation of PSF in the Presence of Jitter

Simulations are performed to demonstrate the proposed jitter-robust phase retrieval algorithms. Here we first discuss how to simulate the PSF in the presence of jitter. We suppose the position of the intersection of the line-of-sight with the image plane at any moment within the exposure time obeys normal distribution. A large series of PSFs with random amount of transverse translation (the orientation is also random) is generated for certain aberration coefficient sets. The values of translation obey normal distribution with a certain value of variance (this value is usually in sub-pixel). These PSFs with different amounts of image translations are superimposed and a new PSF is obtained after normalization. On this basis, a certain amount of noise can be added to this image to simulate the true noisy condition.

Suppose the aperture of the optical system is 2000 mm, the focal length of the system is 28,000 mm, the wavelength is 632.8 nm, the pixel size is 5.5 μm, and the defocusing distance between two PSF images is 1 mm. Three pairs of simulated PSF images in the presence of different jitter parameters for the same wavefront error aberration coefficients and noise level are presented in Figure 4. Figure 4a shows a pair of PSFs without jitter. The magnitude of jitter in Figure 4b–d are 0.1 pixels, 0.3 pixels, and 0.5 pixels, respectively.

We can see from Figure 4 that, as the magnitudes of jitter increases, the details of PSF images become vague. In other words, in the presence of non-ignorable jitter, the intensity distribution of PSF can change a lot. In this case, a considerable error will be introduced when the effects of jitter are not taken into consideration.

### 3.2. Results for One Case and Analysis

Traditional phase retrieval algorithms that do not take jitter into consideration and the two jitter-robust phase retrieval approaches are applied to those three pairs of simulated PSF images. The results are presented in Table 1. In Table 1, C4~C9 represent Fringe Zernike aberration coefficients, and the root mean square error (RMSE) between the true value and recovered value is used to evaluate the accuracy of these algorithms. Here only low-order aberration coefficients are considered, which helps us to know the aberration terms that are sensitive to jitter.

In Table 1, line A1 and line B1 represent the results of the traditional iterative-transform phase retrieval algorithm and parametric phase retrieval algorithm, respectively. Specifically, we use Hybrid input/output (HIO) as the traditional iterative-transform phase retrieval algorithm and we use phase diversity proposed by R. A. Gonsalves as the traditional parametric phase retrieval algorithm. Line A2 and line B2 represent the results of the jitter-robust iterative-transform phase retrieval algorithm and jitter-robust parametric phase retrieval algorithm, respectively.

The following conclusions can be drawn from Table 1:(1)While jitter-robust phase retrieval algorithms cannot thoroughly eliminate the effects of jitter on phase retrieval accuracy, they can effectively reduce its effects and maintain wavefront accuracy.(2)The accuracy of C4 (defocus aberration) and C5, C6 (astigmatism) are more susceptible to the effects of jitter for traditional phase retrieval algorithms.(3)It seems that the accuracy of the parametric jitter-robust phase retrieval algorithm is a little higher than the accuracy of the iterative-transform jitter-robust phase retrieval algorithm.

Then we take the case of sigma = 0.3 pixels as an example to test the convergence of these four algorithms, and the results are presented in Figure 5 (the initial value for these four algorithms are the same as each other). We can see that the parametric algorithms (B1, B2) with an analytic partial derivative of E with respect to an aberration coefficient $\alpha_{j}$ have a higher convergence efficiency, and only 30 times of iteration are needed for convergence. More iterations are needed for iterative-transform algorithms to converge (about 150 times of iteration). Other cases are also tested and similar results are obtained.

### 3.3. Monte Carlo Analysis and Discussion

Monte Carlo simulations are further presented to demonstrate the effectiveness of the proposed jitter-robust approaches. Here the number of aberration coefficients considered is increased to 36. A total of 100 sets of aberration coefficients are randomly generated within the range of [−0.5λ, 0.5λ], and 100 pairs of PSF images are then generated according to the simulation process presented in Section 3.1 and certain system parameters (mainly including aperture size, focal length, pixel size, wavelength, defocusing distance) for a certain jitter parameter. Three jitter parameters (sigma = 0.1 pixels, 0.3 pixels, and 0.5 pixels) and two noise levels (40 dB and 30 dB) are considered for each PSF image (and therefore there are 600 pairs of PSF images in total). The two traditional phase retrieval algorithms (A1, B1) and the two jitter-robust phase retrieval algorithms (A2, B2) are applied to each pair of the PSF images. RMSE of each algorithm are calculated and analyzed. The results are presented in Figure 6.

To simulate noise, we model each image to have Gaussian CCD read noise with a standard deviation of $50e^{-}$ and a dark current of $0.3e^{-}/s$ over a 1 s integration time. The photon noise which is dependent on intensity follows a Poisson distribution.

The peak pixel SNR is defined as
(15)$$\;PSNR=20log_{10}\left(\frac{S_{peak}}{\sqrt{S_{peak}+\sigma_{read}^{2}+\sigma_{dark}^{2}}}\right)\;$$
where Speak is the peak pixel value of the noise-free image, and $\sigma_{read}^{2}$ and$\;\sigma_{dark}^{2}$ are the variances associated with the readout noise and the dark current noise at each pixel, respectively. Simulations will be performed to evaluate the effectiveness of the proposed cross-iteration deconvolution strategy in the noisy conditions. The peak of the PSF is set to 15,000 million photons, which is restricted to full well electron numbers. Then the final peak pixel PSNR is approximately equal to about 40 dB.

The following conclusions can be drawn from Figure 6:(1)We can see from Figure 6a,b that the accuracy of the traditional phase retrieval algorithms decreases sharply as the magnitude of jitter increases, while the two jitter-robust phase retrieval algorithms can reduce the effects of jitter on phase retrieval accuracy.(2)We can see from Figure 6a,b that the RMSE of the jitter-robust phase retrieval algorithms is about half of the RMSE of the traditional phase retrieval algorithms.(3)Comparing Figure 6a,b, we can see that iterative-transform phase retrieval algorithms (both A1 and A2) are more sensitive to noise level, and their accuracy is lower than parametric phase retrieval algorithms for low signal-to-noise-ratio (SNR).

We can see that, compared to traditional phase retrieval algorithms, the accuracy of jitter-robust phase retrieval algorithms is insensitive to the presence of jitter.

## 4. Experiment

Real experiments are performed to demonstrate the effectiveness and practicality of the proposed jitter-robust algorithms. A physical map of the optical path used to generate PSF images including the effects of jitter is shown in Figure 7. Importantly, a hexapod is used to change the pose of a flat mirror, which determines the direction of the optical axis (pointing direction). A large number of pointing change values are randomly generated for a certain magnitude of variance and introduced to the optical system to represent the effect of jitter, and a PSF image is collected each time the pointing is changed (the position and pose of the flat mirror with respect to the hexapod should be carefully determined). By superimposing all these images (100 images), a final PSF image is obtained to represent a real PSF for a certain magnitude of jitter. In this experiment, the variances of the pointing change are selected as 0.1 pixels, 0.3 pixels and 0.5 pixels (this value should be converted to the pitch and torsion angle of the hexapod), respectively. Traditional phase retrieval algorithms and the jitter-robust phase retrieval algorithms are applied to these PSF images.

The results are presented in Figure 8. In this figure, “original images” is obtained by superimposing 100 single images which contain a certain magnitude of image translation (the variance of image translation are 0.1 pixels, 0.3 pixels, and 0.5pixels, respectively). In other words, “original images” represents those images obtained in the presence of certain magnitude of jitter. “Reconstructed images” is calculated with the wavefront phase recovered by each algorithm according to the Fourier optics principle (Gaussian convolution is also considered for the case of A2 and B2). In this experiment, only low-order aberration coefficients are considered and the RMSE of each set of aberration coefficients are presented under the corresponding reconstructed images (we take the phase recovered from a pair of images without jitter as the standard value). Some conclusions can be drawn from Figure 8:(1)The reconstructed images for A2 and B2 are very similar to the original images, and RMSEs for A2 and B2 are smaller than RMSEs for A1 and B1, which demonstrates the effectiveness and practicality of the proposed jitter-robust phase retrieval algorithms.(2)The reconstructed images for A1 are also similar to the original images. However, the RMSE for A1 is quite large. This fact indicates that traditional phase retrieval algorithms attribute image vague to aberration, not jitter. Therefore, while the reconstructed images are similar to the original ones, the accuracy is low.(3)We find that in this experiment, the accuracy of B2 is a litter higher than A2. The possible reason may be that the parametric jitter-robust phase retrieval algorithm is more robust to image noise than the iterative-transform jitter-robust phase retrieval algorithm. This conclusion is consistent with the results presented in Figure 6.

## 5. Other Discussion

In the previous sections, we consider that the magnitude of jitter is known to us. In this section, we further discuss the case where the magnitude of jitter is unknown to us (we suppose that the magnitude of jitter is isotropic). In this case, the iterative jitter-robust phase retrieval algorithms are no longer applicable. For an unknown magnitude of jitter, the procedure of deconvolution in Figure 2 cannot be implemented.

On the other hand, the parametric jitter-robust phase retrieval algorithm presented in Section 2.3 can be extended to the case where the magnitude of the jitter is unknown. In this case, the unknown quantity vector x (which is also the variable of the optimization process) should be rewritten as
(16)$$x^{\prime }=\left[\alpha_{1,}\alpha_{2},\alpha_{3},\cdots \alpha_{N},\sigma \right]$$

In other words, in the optimization process of searching the sets of aberration coefficients, the magnitude of jitter can also be solved. At the same time, aberration coefficients are more accurate since this optimization process has taken the unknown jitter parameter into consideration.

Monte Carlo simulations are also presented to demonstrate the accuracy of the proposed jitter-robust approaches for an unknown magnitude of jitter (i.e., sigma is unknown) (we no longer perform experiments to demonstrate its effectiveness). A total of 100 sets of aberration coefficients are randomly generated within the range of [−0.5λ, 0.5λ], and 100 pairs of PSF images are then generated according to the simulation process presented in Section 3.1 and certain system parameters (mainly including aperture size, focal length, pixel size, wavelength, defocusing distance) for a certain jitter parameter. Three jitter parameters (sigma = 0.1 pixels, 0.3 pixels, and 0.5 pixels) and two noise levels (40 dB and 30 dB) are considered for each PSF image. Comparison between the case where the sigma is known and the case where the sigma is unknown is presented in Figure 9.

Comparing (a) and (b) in Figure 9, we can see that in the case of known and unknown jitter amplitude, the accuracy of the robust phase recovery algorithm with known jitter amplitude is higher than that of the phase recovery algorithm with unknown jitter.

## 6. Conclusions

To conclude, this paper proposes a framework for jitter-robust image-based wavefront sensing algorithm by including the effects of jitter and proposed two kinds of jitter-robust phase retrieval algorithms, which can be categorized into iterative-transform algorithms and parametric algorithms, respectively. The effect of the jitter shows some stochastic properties and it is hard to present an analytic solution to this problem. This paper utilizes two-dimensional Gaussian convolution to describe the effect of jitter on the image, keeping in mind that the variance of this Gaussian function should be consistent with the magnitude of jitter. Further discussions are presented for the cases where the magnitude of jitter is unknown to us, and the parametric jitter-robust phase retrieval algorithm can be extended to this case. Detailed simulations and a real experiment are performed to demonstrate the effectiveness and practicality of the proposed approach.

Overall, we find that the parametric jitter-robust phase retrieval algorithm is superior to the iterative-transform jitter-robust phase retrieval algorithm. This method can be extended to those cases where the magnitude of jitter is unknown to us, which is of significance for the application of the jitter-robust phase retrieval algorithm in practical situations.

## Figures and Tables

**Figure 1 sensors-22-05584-f001:**
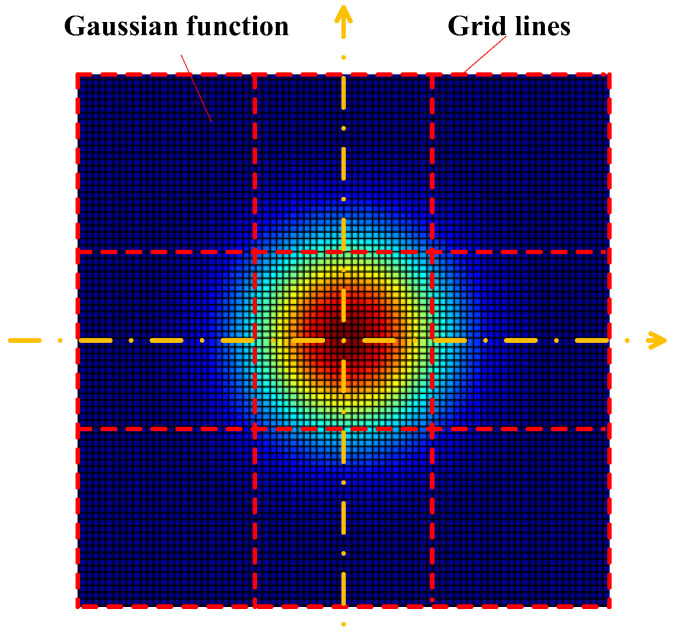
Illustration for obtaining the Gaussian kernel used to describe the effects of jitter. The variance of this Gaussian function should be consistent with the magnitude of jitter.

**Figure 2 sensors-22-05584-f002:**
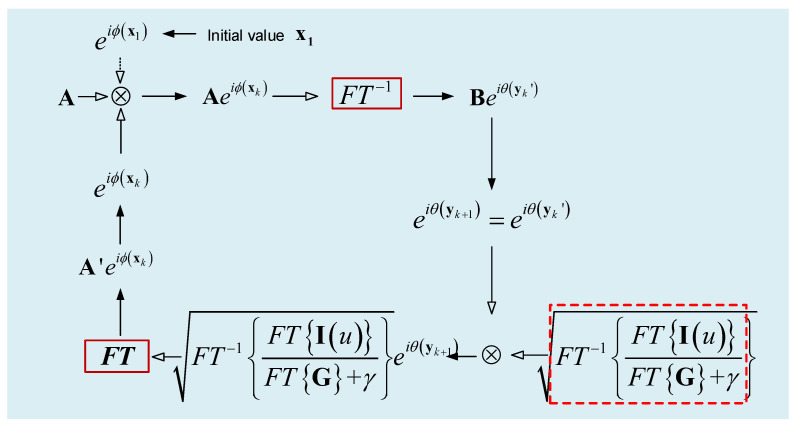
Illustration of the iterative-transform jitter-robust phase retrieval wavefront sensing.

**Figure 3 sensors-22-05584-f003:**
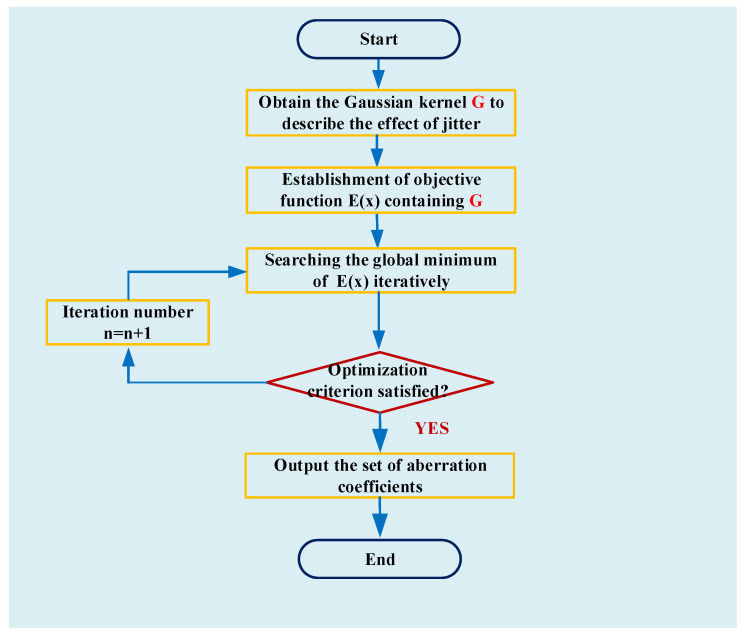
Diagram of parametric jitter-robust phase retrieval approach.

**Figure 4 sensors-22-05584-f004:**
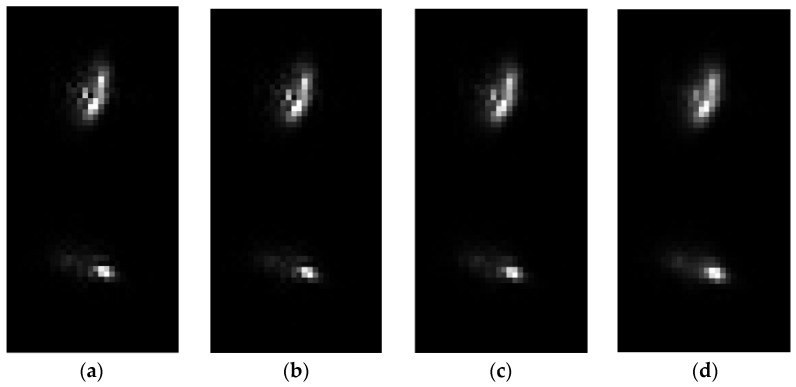
Four pairs of simulated PSF images in the presence of different jitter parameters for the same wavefront error and noise level. (**a**) shows a pair of PSFs without jitter. The magnitudes of jitter in (**b**–**d**) are 0.1 pixels, 0.3 pixels, and 0.5 pixels, respectively. We can see that, as the magnitudes of jitter increases, the PSF images become vague.

**Figure 5 sensors-22-05584-f005:**
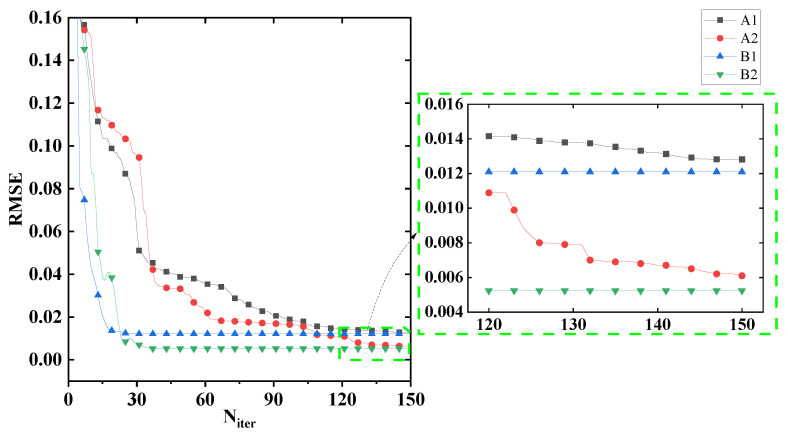
Convergence of the four algorithms for the case of sigma = 0.3 pixels in Table 1. We can see that the parametric algorithms (B1, B2) with an analytic partial derivative of **E** with respect to an aberration coefficient $\alpha_{j}$ have a higher convergence efficiency.

**Figure 6 sensors-22-05584-f006:**
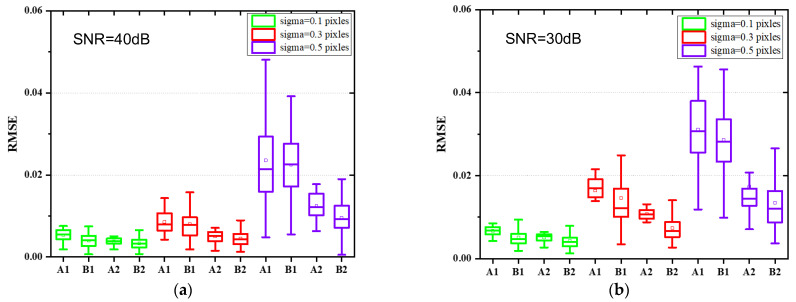
Comparison between the accuracy of two traditional phase retrieval algorithms (A1, B1) and the two jitter-robust phase retrieval algorithms (A2, B2) for a high SNR (**a**) and a low SNR (**b**).

**Figure 7 sensors-22-05584-f007:**
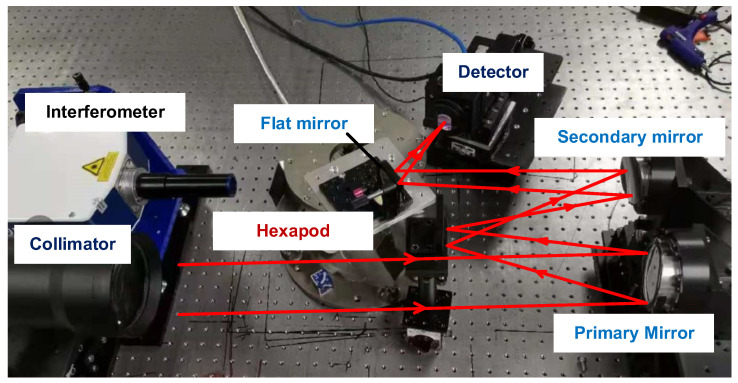
Optical path used to generate PSF images including the effects of jitter.

**Figure 8 sensors-22-05584-f008:**
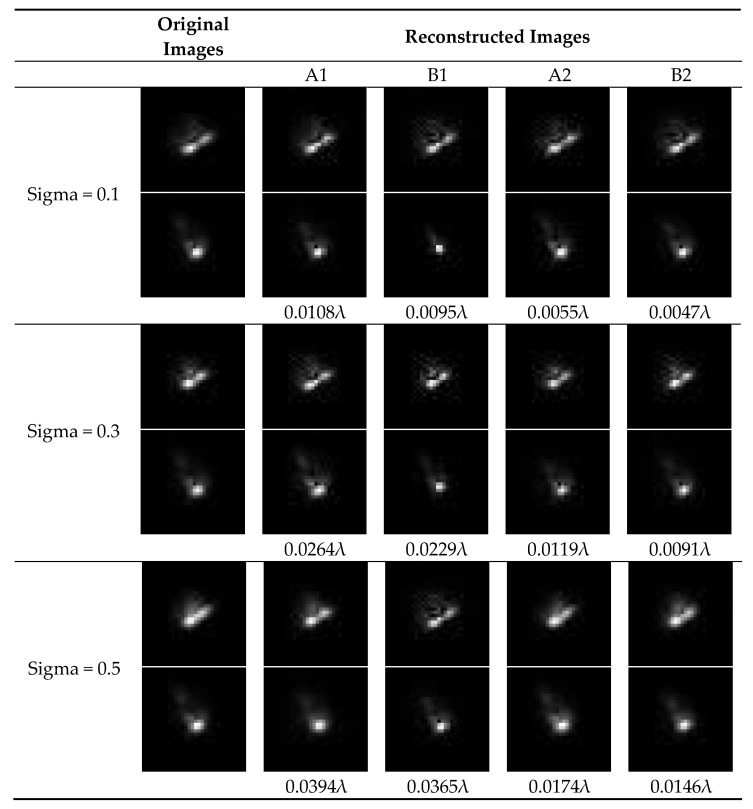
Results of the Traditional phase retrieval algorithms (A1, B1) and the jitter-robust phase retrieval algorithms (A2, B2) when they are applied to those images obtained in the experiment which include the effects of jitter.

**Figure 9 sensors-22-05584-f009:**
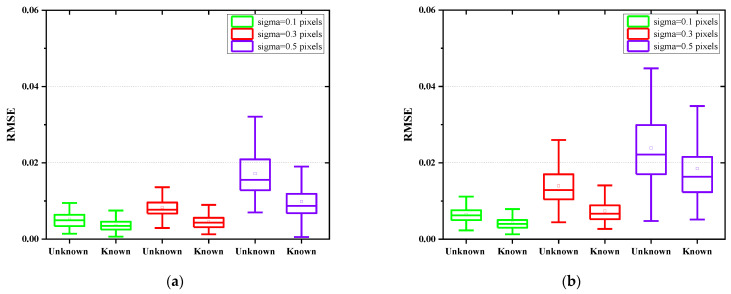
Comparison between the case where the sigma is known and the case where the sigma is unknown for a low SNR (**a**) and a high SNR (**b**). For a high SNR, the accuracy of phase retrieval in the two cases is comparable. For a low SNR, the accuracy of phase retrieval in the case where the sigma is unknown is much lower than in the case where the jitter magnitude is known.

**Table 1 sensors-22-05584-t001:** Comparison of traditional phase retrieval algorithms which do not take jitter into consideration (A1, B1) and the two jitter-robust phase retrieval approaches (A2, B2) for the PSF pairs in Figure 3.

		C4	C5	C6	C7	C8	C9	RMSE	Iterations
	True value	0.321	−0.254	0.223	−0.113	0.036	0.038		
sigma = 0.1 pixels	A1	0.303	−0.250	0.214	−0.114	0.040	0.043	0.0079	142
	B1	0.320	−0.263	0.242	−0.131	0.023	0.031	0.0050	141
	A2	0.319	−0.249	0.223	−0.115	0.040	0.041	0.0031	33
	B2	0.323	−0.258	0.225	−0.115	0.037	0.037	0.0024	29
sigma = 0.3 pixels	A1	0.296	−0.242	0.213	−0.111	0.047	0.045	0.0128	144
	B1	0.334	−0.265	0.245	−0.121	0.027	0.026	0.0121	145
	A2	0.310	−0.245	0.218	−0.110	0.030	0.039	0.0059	34
	B2	0.324	−0.255	0.230	−0.119	0.032	0.035	0.0052	32
sigma = 0.5 pixels	A1	0.287	−0.224	0.205	−0.106	0.044	0.031	0.0217	147
	B1	0.293	−0.216	0.209	−0.111	0.039	0.030	0.0209	145
	A2	0.336	−0.263	0.255	−0.115	0.031	0.035	0.0134	38
	B2	0.327	−0.268	0.249	−0.124	0.033	0.041	0.0119	40

## Data Availability

Not applicable.

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
