# Peer review of "Jitter-Robust Phase Retrieval Wavefront Sensing Algorithms"

_sensors, 2022, doi:10.3390/s22155584_

Round 1
Reviewer 1 Report
I reviewed 'Jitter-robust phase retrieval wavefront sensing' by Guo et al. The manuscript describes a novel algorithm to perform phase diversity in the presence of image jitter. The manuscript shows the algorithm working better than traditional methods in both simulation and experimentally. There are however a number of issues to be resolved before acceptance:
The novel algorithm is compared against 'traditional iterative-transform phase retrieval algorithm and parametric phase retrieval algorithm'. But the specific algorithms are not defined and without knowing these, the results in this manuscript cannot be repeated. For example, is the phase retrieval algorithm Gerchberg-Saxton, Error Reduction, Hybrid input/output, difference map? And are the traditional methods chosen the best to deal with the jitter?
Is Section 5 that deals with the unknown jitter case completed in simulation or experiment? This is not defined in the text.
I find the mathematical notation in the equations incredibly confusing. You have the co-ordinate in the pupil plan as x and then take a Fourier transform to get the intensity which also has co-ordinate of x (although a different font). It would be much clearer if it was x in one domain and say 'u' in the other domain. This is the standard approach.
You have 'G' as the Gaussian in Eq 2 but in Eq 3 you have 'g'. Shouldn't these be the same case?
In Figure 2, you need to define y \theta and k in the text.
You should include a reference for the convolution derivation principle (line 153).
I couldn't see how Eq 12 is derived so maybe you could include an extra step or sentence to help.
2800mm would be better stated as 28m and 2000mm as 2m in my opinion.
Line 193, you say Fringe Zernike coefficients. Are these the same as Zernike coefficients? If not, include a reference. What ordering are you using for the Zernikes? Noll?
How many iterations are used to produce Table 1? Or is it to convergence in each case? What is the convergence criterion?
How is the noise added? Is is photon and/or read noise? What do 30dB and 40dB mean in terms of photon noise?
30dB and 40dB sound similar to me. Why were these numbers chosen?
The paragraph starting Line 272 occurs before Figure 8 is referenced.
Conclusion 3 on Line 295 could easily be tested in Simulation by varying the SNR.
Line 293 I'm not sure 'vague' is the word you want there.
Line 289 and 291 'is' should be 'are' in both cases as the subject is plural.
Figure 9 caption on Line 328 is wrong. This figure does not show a comparison between the sigma is known and sigma is unknown. Figure 9 only shows the unknown case for 2 different SNRs. The last 2 sentences of the caption for Figure 9 should be removed. The figure caption should state what is in the figure. The discussion of the figure should occur in the main body of the paper.
Line 324: it is unclear from the text which is higher.
Line 327: Something is wrong here: "than the case where the case is known" doesn't make sense to me. Do you mean 'jitter magnitude'?
Line 339: Do you mean 'inconsistent' or 'consistent'?
Reviewer 2 Report
The manuscript presents a couple of iterative methods to restore images that have been blurred by platform jitter, like in man made satellites. The manuscript English is poor with several grammatical errors. Lack of articles etc. For example, on line 106 and again 110-111 the authors use the expression "in consistent" when it should be just "consistent". Many more issues that are beyond the review process.
From the technical point of view the authors present a pretty simple blur model based on a Gaussian convolution to produce then an inverse model that has a reduction of blur. The area of jitter blur and blur reduction is a rich area of research with many excellent papers and even book chapters. None of this is mentioned in the references and no attempt is made to put this research effort in the overall area context. No attempt to compare their method with other methods etc.
I strongly suggest that the authors do a much more in depth literature search and get more acquainted with what has already been done in this field.
They can start with the following references for example:
Tao Sun et al., " Application of attitude jitter detection based on short-time asynchronous images and compensation methods for Chinese mapping satellite-1” Optics Express 23(2) 1395-1410 (2015)
Shjie Liu et al., “Dynamic PSF-based jitter compensation and quality improvement for push-broom optical images considering terrain relief and TDI effect” App. Opt. 61(16) 4655 (2022)
Shjie Liu et al., “Attitude jitter detection based on remotely sensed images and dense ground control: A case study for Chinese ZY-3 satellite”, IEEE Journal Select. Topics in Applied Earth Observations and Remote Sensing 9(12) 5760 (2016)
Planetary Remote Sensing and Mapping (book) CRC Press 2018 Chapter on “Correcting Spacecraft jitter in HiRISE images”
Round 2
Reviewer 1 Report
I read the authors' reply to my review of 'Jitter-robust phase retrieval wavefront sensing' by Guo et al. The authors have addressed each point and I'm happy with most of them. Some I don't understand or don't agree with their comments, and should be further addressed (or the editor should decide on whether the paper should be accepted or not now). I will go through the numbered points from the authors' reply:
1. ok
2. ok
3. I don't think the authors understood my main point here. If you take the Fourier transform of a variable in the x domain (\phi(x) in Equation 1) you must then use another variable (eg u) in the other Fourier domain. So I(x) should be I(u). Otherwise you have the x variable in both Fourier domains. The two x variables are slightly different fonts but this is confusing to the reader.
Also note that g is now missing from Equation 3 in the paper.
4. The sentence added to the paper "g is the Gaussian function which used to discuss the property of Gaussian function" doesn't make sense to me. It seems a circular definition to me.
5. ok
6. ok
7. ok but an extra line in the paper would help
8. ok
9. ok but I still think you should add a reference to Fringe Zernikes
10. You haven't really answered the question. You should make an explicit sentence in the paper about how many iterations are used or what the convergence criterion is when you produce Table 1.
11. You should state how much photon and readout/dark noise there is in each case.
12. ok
13. ok
14. ok
15 ok
16. ok
17. Note that copy and pasting the caption from Word with author track changes on to your reply document also copies the deleted text. You say high low SNR (a) in the reply which is well confusing. The paper looks ok.
However, by changing Figure 9 you seem to have caused the text to become out of date. On line 346, you say "Comparing Fig. 9 and Fig. 6" but you don't need to refer back to Fig 6 now you have all the information in Figure 9.
18. ok
19. ok
20. ok
Reviewer 2 Report
I appreciate the effort the authors put in revising the manuscript. However, i strongly feel the the revision fall short of what is needed to make this manuscript publishable in an archival journal.
